# Griffithsin and Carrageenan Combination Results in Antiviral Synergy against SARS-CoV-1 and 2 in a Pseudoviral Model

**DOI:** 10.3390/md19080418

**Published:** 2021-07-26

**Authors:** Sahar Alsaidi, Nadjet Cornejal, Oneil Mahoney, Claudia Melo, Neeharika Verma, Thierry Bonnaire, Theresa Chang, Barry R. O’Keefe, James Sailer, Thomas M. Zydowsky, Natalia Teleshova, José A. Fernández Romero

**Affiliations:** 1Population Council, New York, NY 10065, USA; sahar.alsaidi@lc.cuny.edu (S.A.); nadjet.cornejal96@bcmail.cuny.edu (N.C.); oneil.mahoney@stu.bmcc.cuny.edu (O.M.); claudia.melo24@bcmail.cuny.edu (C.M.); nverma@popcouncil.org (N.V.); tbonnaire@popcouncil.org (T.B.); jsailer@popcouncil.org (J.S.); tzydowsky@popcouncil.org (T.M.Z.); nteleshova@popcouncil.org (N.T.); 2Department of Science, Borough of Manhattan Community College, The City University of New York, New York, NY 10007, USA; 3Department of Anthropology, Lehman College, The City University of New York, New York, NY 10468, USA; 4Center for Achievement in Science Education, Department of Biology and Chemistry, School of Natural and Behavioral Sciences, Brooklyn College, The City University of New York, New York, NY 11210, USA; 5Department of Microbiology, Biochemistry and Molecular Genetics, New Jersey Medical School, Rutgers, The State University of New Jersey, Newark, NJ 07102, USA; changth@njms.rutgers.edu; 6Natural Products Branch, Molecular Targets Program, Developmental Therapeutics Program, Center for Cancer Research, Division of Cancer Treatment and Diagnosis, National Cancer Institute, Frederick, MD 21702, USA; okeefeba@mail.nih.gov

**Keywords:** SARS-CoV-1, SARS-CoV-2, lectins, sulfated polysaccharides

## Abstract

Over 182 million confirmed cases of COVID-19 and more than 4 million deaths have been reported to date around the world. It is essential to identify broad-spectrum antiviral agents that may prevent or treat infections by severe acute respiratory syndrome coronavirus 2 (SARS-CoV-2) but also by other coronaviruses that may jump the species barrier in the future. We evaluated the antiviral selectivity of griffithsin and sulfated and non-sulfated polysaccharides against SARS-CoV-1 and SARS-CoV-2 using a cytotoxicity assay and a cell-based pseudoviral model. The half-maximal cytotoxic concentration (CC_50_) and half-maximal effective concentration (EC_50_) were determined for each compound, using a dose-response-inhibition analysis on GraphPad Prism v9.0.2 software (San Diego, CA, USA). The therapeutic index (TI = CC_50_/EC_50_) was calculated for each compound. The potential synergistic, additive, or antagonistic effect of different compound combinations was determined by CalcuSyn v1 software (Biosoft, Cambridge, UK), which estimated the combination index (CI) values. Iota and lambda carrageenan showed the most potent antiviral activity (EC_50_ between 3.2 and 7.5 µg/mL). Carrageenan and griffithsin combinations exhibited synergistic activity (EC_50_ between 0.2 and 3.8 µg/mL; combination index <1), including against recent SARS-CoV-2 mutations. The griffithsin and carrageenan combination is a promising candidate to prevent or treat infections by SARS-CoV-1 and SARS-CoV-2.

## 1. Introduction

The emergence of a novel coronavirus, quickly escalated into the current pandemic that was declared on 11 March 2020, by the World Health Organization. The pandemic has placed healthcare systems under extreme pressure; globally, over 182 million confirmed cases of COVID-19 and more than 4 million deaths have been reported to date [1]. The novel virus, severe acute respiratory syndrome coronavirus 2 (SARS-CoV-2), belongs to the *Coronaviridae* family. This family includes SARS-CoV-1 and MERS-CoV, two zoonotic viruses that emerged in 2003 and 2012, respectively [2]. Although highly effective vaccines targeting SARS-CoV-2 spike proteins have been approved for emergency use by multiple stringent regulatory authorities, accumulation of mutations in the spike protein may allow the virus to be transmitted more effectively and, in the worst-case scenario, evade the immune response triggered by vaccines [3,4,5,6]. 

The coronaviruses have shown an impressive ability to jump the species barrier, and more coronaviruses may migrate from animal reservoirs to humans in the future. For these reasons, it is essential to identify broad-spectrum antiviral agents that inhibit different coronaviruses and help to prevent/treat the diseases caused by these viruses. SARS-CoV-1 and SARS-CoV-2 have a similar natural history of infection; both enter the upper respiratory tract and infect the epithelial cells lining the respiratory tract. They enter target cells by binding of their surface spike proteins to human angiotensin-converting enzyme 2 (ACE2), the primary viral receptor, present on the surface of target cells [7]. Therefore, antiviral agents that target SARS-CoV-1 and SARS-CoV-2 entry have the potential to prevent/treat these infections. 

Naturally occurring antiviral agents, such as griffithsin (GRFT, Figure 1A), an antiviral lectin, and carrageenan (CG, Figure 1B), a sulfated polysaccharide, can target a wide range of enveloped viruses [8]. Importantly, GRFT has previously been reported to have broad-spectrum in vitro activity against *Coronaviridae* and in vivo activity against SARS-CoV-1 in a mouse model system following intranasal administration [9]. Similarly, the activity of sulfated polysaccharides against SARS-CoV-2 is documented in the literature [10,11]. 

GRFT is a homodimeric lectin of 121 aminoacids and six carbohydrate-binding sites with high affinity for high mannose arrays. These mannose arrays are frequently found in viral spikes of important pathogens like HIV, HSV, hepatitis C virus (HCV), ebola virus, and members of the *Coronaviridae* family [13]. The ability to block HIV in vitro, at picomolar concentrations, makes GRFT one of the most potent molecules inhibiting HIV replication [14]. Although EC_50_ values are higher against coronaviruses, GRFT is potent enough to make this naturally occurring agent a promising candidate to fight the current SARS-CoV-2 pandemic. In addition to the drug potency, GRFT is poorly immunogenic, and several studies in animal models have shown its excellent safety profile [15]. CGs are polysaccharides isolated from red seaweed; its broad antiviral spectrum includes viruses like HSV, rhinoviruses, and coronaviruses. The antiviral activity is well-documented against HPV and is probably the most potent anti-HPV agent reported in the literature [16,17]. Several preclinical and clinical studies have shown its excellent safety profile after vaginal and respiratory routes of administration [18,19]. Additionally, CG is Generally Recognized as Safe (GRAS) by the Food and Drug Administration and is a common food additive [20]. Herein, we further explore the potential antiviral selectivity of GRFT, non-sulfated and sulfated polysaccharides, and combinations thereof against SARS-CoV-1 and SARS-CoV-2.

## 2. Results

Table 1 shows the half-maximal cytotoxic concentration (CC_50_), half-maximal effective concentration (EC_50_), and therapeutic index (TI) values for all the compounds tested. GRFT showed broad-spectrum antiviral activity against SARS-CoV-1 and 2. While the polysaccharides with selective antiviral activity were all sulfated, the sulfate group’s presence is not a determining factor of the efficacy; some sulfated polysaccharides, such as heparin, heparan sulfate, fucoidan, and chondroitin sulfate did not show antiviral activity against SARS-CoV-1 or SARS-CoV-2. Among the polysaccharides, ι-CG and λ-CG had the best activity, with EC_50_ values below 7.5 µg/mL. Therefore, we focused on ι-CG and λ-CG, evaluating their combination with GRFT. Previous studies have shown the GRFT and CG combination results in synergistic or additive effects against different viruses including HPV and HSV [8]. The potential synergistic effect of the GRFT and CG combination against SARS-CoV-1 and SARS-CoV-2 was determined by CalcuSyn software (Biosoft, Cambridge), which estimated the combination index (CI) values. As shown in Table 2, ι-CG and λ-CG inhibited entry of SARS-CoV-1 and all SARS-CoV-2 PsVs with lower EC_50_ values (between 3.2 and 7.5 µg/mL) than those shown by GRFT (between 12.5 and 37.6 µg/mL). The EC_50_ of GRFT against all three SARS-CoV-2 PsVs was larger when compared to SARS-CoV-1 PsV. However, some of the most relevant mutations recently identified in the SARS-CoV-2 spike protein (D614G, K1417N/E484K/N501Y), did not significantly impact the susceptibility of these PsVs to GRFT. Interestingly, GRFT/CG combinations showed more potent antiviral activity, specially at a 5:1 ratio, as indicated by the lower EC_50_ values in Table 2. The data also revealed their synergistic activity, based on CI values below 1. 

## 3. Discussions

This synergistic in vitro activity of GRFT and CG suggests that formulations delivering combinations of GRFT and CG might be useful in preventing or treating infections caused by SARS-CoV-1 or SARS-CoV-2. In addition to the potent activity of GRFT and CG against SARS-CoV-2, CG has been shown to have activity against the common cold [21], which is caused by a variety of respiratory viruses, including coronaviruses. Clinical studies show significant reduction in severity of common cold symptoms in subjects that administered CG nasally. Furthermore, data in mice inoculated intranasally with influenza A virus in the presence of ι-CG, demonstrated potent activity of CG against influenza. This activity is mediated through direct binding of CG to viral particles, which inhibits viral adsorption and internalization [22]. 

Importantly, in vivo studies will still be required to confirm this in vitro synergistic activity in animal models. Pharmacokinetic and pharmacodynamic studies will be essential to understand the potential of this combination [23]. Our formulation will require topical administration through the upper respiratory tract contrary to other antivirals currently used to treat SARS-CoV-2 infections that require intravenous (IV) administration. One of these antivirals is remdesivir, an inhibitor of the viral RNA-dependent RNA polymerase (RdRp) that requires daily IV doses of 100 or 200 mg [24]. The GRFT and CG combination has the advantage of using two naturally occurring agents whose safety has been thoroughly studied in different animal models and clinical studies [13,15,18,25,26]. The mode of action of these molecules does not require systemic adsorption, and topical application in the upper respiratory tract will interfere with the coronaviruses’ attachment and entry to epithelial cells.

## 4. Conclusions

These results strongly suggest that a GRFT/CG combination could provide broad-spectrum antiviral activity targeting different respiratory viruses, including a range of coronaviruses. The extensive preclinical safety data available for the GRFT and CG combination [8,25] and the results shown herein support testing this combination in preclinical efficacy and toxicology models and its subsequent clinical evaluation for delivery in the upper (and possibly lower) respiratory tract.

## 5. Methods

### 5.1. Cell Line and Pseudovirus Production

The human angiotensin-converting enzyme 2 (hACE-2)-expressing HeLa cells (HeLa ACE-2) were provided by Dr. Dennis Burton (The Scripps Research Institute, La Jolla, CA, USA). SARS-CoV-1 and SARS-CoV-2 pseudoviruses (PsV) were produced following the procedure described by Schmidt et al., 2020 [27]. Plasmids containing the SARS-CoV-1 or SARS-CoV-2 spike genes [pSARS-CoV1-Strunc, pSARS-CoV2-Strunc (original Wuhan strain), pSARS-CoV2-Strunc (D614G mutation), pSARS-CoV2-Strunc (K1417N/E484K/N501Y mutations)], pCRV1NHG GagPol, and pNanoLuc2AEGFP were used to produce the PsVs [27]. pCRV1NHG GagPol, pNanoLuc2AEGFP, and pSARS-CoV1-Strunc or pSARS-CoV2-Strunc plasmids were used to transfect 293T cells (ATCC, Manassas, VA, USA) seeded in 6-well plates, using lipofectamine 2000 (ThermoFisher Scientific, Waltham, MA, USA). The DNA/lipofectamine 2000 mixtures were added to 293T cell monolayers and incubated for 6 h at 37 °C, 5% CO_2_, and 98% humidity. After this brief incubation, the cell monolayers were washed twice with D-PBS (ThermoFisher Scientific, Waltham, MA, USA), and DMEM (ThermoFisher Scientific, Waltham, MA, USA) with 10% FBS (ThermoFisher Scientific, Waltham, MA, USA) and Penicillin + Streptomycin (ThermoFisher Scientific, Waltham, MA, USA) was added to each well. The plates were incubated for 48 h at 37 °C, 5% CO_2_, 98% humidity, and the cell supernatants were collected, filtered (using a 0.22 μm pore size PVDF filter), aliquoted and stored at −80 °C. The viral titers (SARS-CoV-1 PsV, SARS-CoV-2 Wuhan PsV, SARS-CoV-2 D614G PsV, and SARS-CoV-2 K1417N/E484K/N501Y PsV) were determined using a cell-based pseudoviral entry assay [27] and the TurboLuc™ Luciferase One-Step Glow Assay Kit (ThermoFisher Scientific, Waltham, MA, USA).

### 5.2. Cytotoxicity and Antiviral Assay

The cytotoxicity and antiviral activity were determined using the XTT [26] and the cell-based pseudoviral entry assay, respectively. Briefly, different concentrations of each compound (Table 1) were added in triplicate to HeLa ACE-2 cells seeded in clear bottom 96-well microplates and then incubated at 37 °C, 5% CO_2_, and 98% humidity for 72 h. Tween 20 (Sigma Aldrich, St. Louis, MO, USA) was used as a positive control for cytotoxicity. XTT (ThermoFisher Scientific, Waltham, MA, USA) was added to all wells after 72-h incubation, and the absorbance was measured at 450 nm using a Spectramax iD3 (Molecular Devices, San Jose, CA, USA). The antiviral activity of the same compounds tested in the XTT assay was evaluated using the cell-based pseudoviral entry assay with SARS-CoV-1 PsV and SARS-CoV-2 Wuhan PsV. The same concentrations/replicates of each compound tested in the XTT assay were challenged (co-treatment) with SARS-CoV-1 PsV or SARS-CoV-2 Wuhan PsV in HeLa ACE-2 cell monolayers seeded in white opaque 96-well microplates. The plates were incubated at 37 °C, 5% CO_2_, and 98% humidity for 72 h. The TurboLuc™ Luciferase One-Step Glow Assay (ThermoFisher Scientific, Waltham, MA, USA) was used to determine the percentage entry of the PsVs in the presence of each compound concentration versus the virus control. The half-maximal cytotoxic concentration (CC_50_) and half-maximal effective concentration (EC_50_) were determined for each compound, using a dose-response-inhibition analysis on GraphPad Prism v9.0.2 software (San Diego, CA, USA). The therapeutic index (TI = CC_50_/EC_50_) was calculated for each compound. 

### 5.3. Combination Studies

The potential synergistic, additive, or antagonistic effect of different compounds combinations was determined by CalcuSyn v1 software (Biosoft, Cambridge, UK), which estimated the combination index (CI) values. For this purpose, equipotential combinations of GRFT and CG were evaluated in this experiment. Based on the EC_50_ values of each compound, ratios of 1:3 or 1:5 (CG:GRFT) were determined as optimal. Each compound alone or in combination was tested against SARS-CoV-1 PsV, SARS-CoV-2 Wuhan PsV, SARS-CoV-2 D614G PsV, and SARS-CoV-2 K1417N/E484K/N501Y PsV. The cytotoxicity and antiviral assay described in the previous section were used to evaluate the toxicity and efficacy of these combinations. The inhibitory effects for each drug alone or in combination were transferred to CalcuSyn software to estimate CI values. All compounds and combinations were tested in at least two independent experiments.

## Figures and Tables

**Figure 1 marinedrugs-19-00418-f001:**
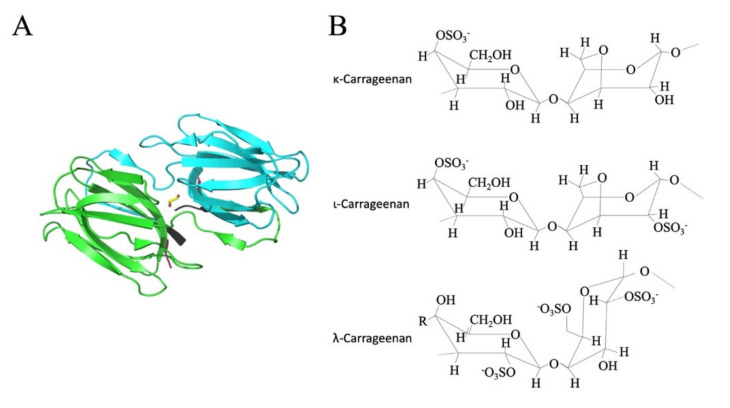
The structure of GRFT and CG. (**A**) The homodimeric native structure of GRFT (adapted from Moulaei et al. [12]). (**B**) The sulfated polysaccharides κ-Carrageenan (κ-CG), ι-Carrageenan (ι-CG), and λ-Carrageenan (λ-CG).

**Table 1 marinedrugs-19-00418-t001:** Selective antiviral activity of GRFT, sulfated and non-sulfated polysaccharides against SARS-CoV-1 and SARS-CoV-2 Wuhan PsVs.

Compounds	Category	Molecular Weight (kDa)	Source	CC_50_ ^†^ (µg/mL)	EC_50_ ^¶^ (µg/mL)		TI *	
					SARS-1 ^§^	SARS-2 ^§^	SARS-1 ^§^	SARS-2 ^§^
Griffithsin (GRFT)	Lectin	12.77	Custom manufactured [14]	>600	12.5	20.6	>48	>29
λ-Carrageenan (λ-CG)	Sulfated polysaccharide	600–700	Gelymar (Santiago de Chile, Chile)	>600	4.2	6.1	>143	>98
ι-Carrageenan (ι-CG)	Sulfated polysaccharide	400–560	Gelymar (Santiago de Chile, Chile)	>600	4.3	7.5	>139	>80
κ-Carrageenan (κ-CG)	Sulfated polysaccharide	400–560	Sigma Aldrich (St. Louis, MO, USA)	>600	8.4	24.2	>71	>25
Dextran sulfate	Sulfated polysaccharide	~15	Sigma Aldrich (St. Louis, MO, USA)	>600	7.8	19.4	>77	>31
Lignosulfonic acid	Polar lignin-derived	52	Beantown Chemical Corp. (Hudson, NH, USA)	>600	93.9	184.4	>6	>3
Heparan sulfate	Sulfated polysaccharide	~30	Sigma Aldrich (St. Louis, MO, USA)	>600	>600	>600	ND	ND
Chondroitin sulfate	Sulfated polysaccharide	14–26	Sigma Aldrich (St. Louis, MO, USA)	>600	>600	>600	ND	ND
Heparin	Sulfated polysaccharide	~15	Sigma Aldrich (St. Louis, MO, USA)	>600	>600	>600	ND	ND
Fucoidan	Sulfated polysaccharide	15–30	Sigma Aldrich (St. Louis, MO, USA)	>600	>600	>600	ND	ND
Mannan	Non-sulfated polysaccharide	34–62	Sigma Aldrich (St. Louis, MO, USA)	>600	>600	>600	ND	ND
Chitosan	Non-sulfated polysaccharide	190–310	Sigma Aldrich (St. Louis, MO, USA)	>600	>600	>600	ND	ND
Carboxymethylcellulose	Non-sulfated polysaccharide	~90	Sigma Aldrich (St. Louis, MO, USA)	>600	>600	>600	ND	ND
Xanthan gum	Non-sulfated polysaccharide	2 × 10^3^–2 × 10^4^	Sigma Aldrich (St. Louis, MO, USA)	>600	>600	>600	ND	ND

^†^ CC_50_: half-maximal cytotoxic concentration; ^¶^ EC_50_: half-maximal effective concentration; * therapeutic index (TI) = CC_50_/EC_50_; ^§^ SARS-CoV-1 and SARS-CoV-2 Wuhan PsVs; ND: not determined.

**Table 2 marinedrugs-19-00418-t002:** Synergistic activity of GRFT and CG against SARS-CoV-1 and SARS-CoV-2 PsVs.

Compounds		EC_50_ ^¶^ (µg/mL)			
		SARS-CoV-1	SARS-CoV-2 Wuhan	SARS-CoV-2 D614G	SARS-CoV-2 K1417N/E484K/N501Y
GRFT		12.5	20.6	37.6	31.6
ι-CG		4.3	7.5	6.5	6.7
λ-CG		4.2	6.1	4.5	3.2
ι-CG + GRFT 1:3 ratio		3.8	2.5	2.7	1.7
λ-CG + GRFT 1:3 ratio		0.7	3.2	2.8	1.9
ι-CG + GRFT 1:5 ratio		0.2	0.7	0.5	0.4
λ-CG + GRFT 1:5 ratio		0.4	0.4	0.6	0.3
**Combination**		**CI values ***			
		**SARS-CoV-1**	**SARS-CoV-2 Wuhan**	**SARS-CoV-2 D614G**	**SARS-CoV-2 K1417N/E484K/N501Y**
ι-CG + GRFT	ED_50_	ND	ND	0.19910	0.28442
	ED_75_	ND	ND	0.11192	0.19067
	ED_90_	ND	ND	0.07208	0.13426
λ-CG + GRFT	ED_50_	0.61385	0.66908	0.35799	0.62309
	ED_75_	0.30337	0.51784	0.25653	0.43368
	ED_90_	0.18899	0.40843	0.19111	0.30924

^¶^ EC_50_: half-maximal effective concentration; * combination index (CI): a quantitative measure of the degree of drug interaction in terms of additive effect (CI = 1), synergism (CI < 1), or antagonism (CI > 1) is shown for a given end point of the effect measurement; ND: not determined.

## Data Availability

Not applicable.

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
