# Peer review of "Griffithsin and Carrageenan Combination Results in Antiviral Synergy against SARS-CoV-1 and 2 in a Pseudoviral Model"

_marinedrugs, 2021, doi:10.3390/md19080418_

Round 1

Reviewer 1 Report

The manuscript describes a screening of several natural compounds (proteins and polysaccharides) for their anti-viral activity. The study relies on the evaluation of the antiviral capacities of these natural compounds by the use of pseudoviruses presenting the SARS-CoV 1 and 2 spike protein at their surface, and HeLa cells expressing the human ACE-2 receptor. The results of EC50 determination highlight the potent inhibition potential of a combination of Griffithsin and carrageenan. The article is very well written, organized and documented. It fits perfectly the scope of Marine Drugs.

Below, I listed some remarks which need to be addressed by the authors before publication.

1- This study, though very convincing, relies on the use of a model of virus that intends to mimic the natural infection without the requirement of a high security level environment. The use of pseudoviruses is a well-documented, robust and validated method for inhibition assays. However, pseudoviruses are not composed of all viral elements. Therefore, I suggest to modify the title accordingly. Indeed, this study probably demonstrates the ability of the compounds tested to inhibit the binding of spike proteins to the ACE-2 but no full SARS-CoV was used.

2- The title of Ref [6] was found to be: “Detection of a SARS-CoV-2 variant of concern in South Africa”. Nature. 2021 Apr;592(7854):438-443. doi: 10.1038/s41586-021-03402-9. Epub 2021 Mar 9. PMID: 33690265.:

3- Line 59: the ref [9] is not about the cited viruses but about coronaviruses.

4- Although the synergistic effect of GRFT and CG is demonstrated, an explanation or hypothesis about the molecular/cellular mechanism of inhibition on SARS-CoV by the combination of compounds would be interesting.

5- It is not clear if the challenge of compounds with the PsV was performed before incubation with the cells or simultaneously. In other words, what kind of time-of-addition assay was performed: virus pre-treatment with compounds (if so, what was the time of incubation before addition to the cells?) or co-treatment of cells and virus with compounds?

6- As many parameters could modify antiviral properties of polysaccharides (degree of sulfation, molecular weight, structural features..) information about characteristics of the PS used should be provided, at least their MW (or the range provided by Sigma Aldrich).

Author Response

Response to reviewers’ comments:

We thank the reviewer for the constructive criticisms. We agree with all the comments and have made the corresponding edits. Below, embedded in the questions, are our answers.

  • This study, though very convincing, relies on the use of a model of virus that intends to mimic the natural infection without the requirement of a high security level environment. The use of pseudoviruses is a well-documented, robust and validated method for inhibition assays. However, pseudoviruses are not composed of all viral elements. Therefore, I suggest to modify the title accordingly. Indeed, this study probably demonstrates the ability of the compounds tested to inhibit the binding of spike proteins to the ACE-2 but no full SARS-CoV was used.

The title has been modified:

Griffithsin and Carrageenan Combination Results in Antiviral Synergy Against SARS-CoV-1 and 2 in a Pseudoviral Model

  • The title of Ref [6] was found to be: “Detection of a SARS-CoV-2 variant of concern in South Africa”. Nature. 2021 Apr;592(7854):438-443. doi: 10.1038/s41586-021-03402-9. Epub 2021 Mar 9. PMID: 33690265.

The title was updated accordingly.

  • Line 59: the ref [9] is not about the cited viruses but about coronaviruses.

The mistake has been corrected. Reference 8 is the only one listed now.

  • Although the synergistic effect of GRFT and CG is demonstrated, an explanation or hypothesis about the molecular/cellular mechanism of inhibition on SARS-CoV by the combination of compounds would be interesting.

We don’t have a concrete answer for this question. We know both compounds interact with the viral spike and prevent viral attachment/entry, but the molecular basis of the synergistic mechanism is unclear.

  • It is not clear if the challenge of compounds with the PsV was performed before incubation with the cells or simultaneously. In other words, what kind of time-of-addition assay was performed: virus pre-treatment with compounds (if so, what was the time of incubation before addition to the cells?) or co-treatment of cells and virus with compounds?

It was a co-treatment. It has been added to the manuscript text.

  • As many parameters could modify antiviral properties of polysaccharides (degree of sulfation, molecular weight, structural features.) information about characteristics of the PS used should be provided, at least their MW (or the range provided by Sigma Aldrich).

A column has been added to the table to show the MW.

Reviewer 2 Report

Sahar Alsaidi et al reports Antiviral 2 Synergy of Griffithsin and Carrageenan Combination against SARS-CoV-1 and 2. After close evaluation of the paper I suggest revision according to the next points:

  1. Please fit the abstract according to the guidelines (one paragraph, number of words)
  2. In abstract: please provide EC50 for carrageenan and griffithsin combinations.
  3. The introduction is focuse\d primary on SARS-CoV. Please extend the introduction with additional information about Griffithsin and Carrageenan. Please explain why the authors select this combination for future testing. Please explain why you expect synergy of these compounds?
  4. In Sect. 2 please explain all abbreviations (line 67) when first time mentioned (CC50, EC50, TI
  5. Please split section 2 for Results and discussion. There is a mix of own results and literature data. It is very difficult to follow.
  6. The tables must be self-explanatory. Please explain all abbreviations in the legend to Tables 1 and 2
  7. In Table 2: why only combinations of the 1:3 ratio and 1:5 ratio were tested. Please justify.
  8. The discussion is very weak. Please compare the results of your study with other natural candidates to cure SARS CoV.
  9. 9. Not always in vitro results are correlated with in vivo. One of the important factors is pharmacokinetic. Please discuss this aspect. The article https://doi.org/10.3390/md18110557 will be helpful.
  10. Please split section 3 for subsection and describe all methods according to subsection.
  11. Please provide structures of Griffithsin and Carrageenan and characterize these compounds (MW, type of carrageenan, etc). How Griffithsin was synthesised (isolated). Please describe.
  12. For synergy calculations see details in https://doi.org/10.1016/j.synres.2018.04.001
  13. Please provide the conclusion.

Author Response

We thank the reviewer for the comments aiming to improve the quality of our manuscript.

Sahar Alsaidi et al reports Antiviral 2 Synergy of Griffithsin and Carrageenan Combination against SARS-CoV-1 and 2. After close evaluation of the paper I suggest revision according to the next points:

  1. Please fit the abstract according to the guidelines (one paragraph, number of words)

The abstract is now one paragraph, and the word count is appropriate.

  1. In abstract: please provide EC50 for carrageenan and griffithsin combinations.

The EC50s are now included.

  1. The introduction is focuse\d primary on SARS-CoV. Please extend the introduction with additional information about Griffithsin and Carrageenan. Please explain why the authors select this combination for future testing. Please explain why you expect synergy of these compounds?

I have included more information about griffithsin and carrageenan.

The following sentence has been included in the methods section to explain the choice of combinations: “Equipotential combinations of GRFT and CG were evaluated in this experiment”. This is the standard approach to study the effect of drugs combinations.

  1. In Sect. 2 please explain all abbreviations (line 67) when first time mentioned (CC50, EC50, TI

The abbreviations are now explained in that section.

  1. Please split section 2 for Results and discussion. There is a mix of own results and literature data. It is very difficult to follow.

The Results and Discussion sections are now separated.

The tables must be self-explanatory. Please explain all abbreviations in the legend to Tables 1 and 2

The abbreviations are now explained in the legend.

  1. In Table 2: why only combinations of the 1:3 ratio and 1:5 ratio were tested. Please justify.

The following sentence has been included in the methods section to explain the choice of combinations: “Equipotential combinations of GRFT and CG were evaluated in this experiment”. This is the standard approach to study the effect of drugs combinations.

  1. The discussion is very weak. Please compare the results of your study with other natural candidates to cure SARS CoV.

I have added a brief paragraph comparing the potential mode of action and route of administration of this combination versus remdesivir.

  1. Not always in vitro results are correlated with in vivo. One of the important factors is pharmacokinetic. Please discuss this aspect. The article https://doi.org/10.3390/md18110557 will be helpful.

I have added a brief comment regarding this issue and your reference has been included.

  1. Please split section 3 for subsection and describe all methods according to subsection.

Section 3 has been divided in subsections as suggested.

  1. Please provide structures of Griffithsin and Carrageenan and characterize these compounds (MW, type of carrageenan, etc). How Griffithsin was synthesised (isolated). Please describe.

The table 1 includes a reference about how GRFT was produced and the MW for all compounds is now included in table 1.

  1. For synergy calculations see details in https://doi.org/10.1016/j.synres.2018.04.001

Thank you, we are aware of this article. Our calculations were made according to the guidelines of the software.

  1. Please provide the conclusion.

There is a conclusion in the resubmitted article.

Round 2

Reviewer 2 Report

The manuscriot could be accepted in the present revised form.